# Conductive Nanosheets Fabricated from Au Nanoparticles on Aqueous Metal Solutions under UV Irradiation

**DOI:** 10.3390/ma17040842

**Published:** 2024-02-09

**Authors:** Maho Tagawa, Hiroto Kaneki, Takeshi Kawai

**Affiliations:** Department of Industrial Chemistry, Tokyo University of Science, 6-3-1 Niijuku, Katsushika-ku, Tokyo 125-8585, Japan

**Keywords:** Au nanoparticles, UV irradiation, conductive nanosheets, air–water interface

## Abstract

Highly transparent, conductive nanosheets are extremely attractive for advanced opto-electronic applications. Previously, we have demonstrated that transparent, conductive Au nanosheets can be prepared by UV irradiation of Au nanoparticle (AuNP) monolayers spread on water, which serves as the subphase. However, thick Au nanosheets cannot be fabricated because the method is not applicable to large Au NPs. Further, in order to fabricate nanosheets with different thicknesses and compositions, it is necessary to prepare the appropriate NPs. A strategy is needed to produce nanosheets with different thicknesses and compositions from a single type of metal NP monolayer. In this study, we show that this UV irradiation technique can easily be extended as a nanosheet modification method by using subphases containing metal ions. UV irradiation of 4.7 nm AuNP monolayers on 480 µM HAuCl_4_ solution increased the thickness of Au nanosheets from 3.5 nm to 36.5 nm, which improved conductivity, but reduced transparency. On the other hand, the use of aqueous AgNO_3_ and CH_3_COOAg solutions yielded Au-Ag hybrid nanosheets; however, their morphologies depended on the electrolytes used. In Au-Ag nanosheets prepared on aqueous 500 µM AgNO_3_, Au and Ag metals are homogeneously distributed throughout the nanosheet. On the other hand, in Au-Ag nanosheets prepared on aqueous 500 µM CH_3_COOAg, AuNPs still remained and these AuNPs were covered with a Ag nanosheet. Further, these Au-Ag hybrid nanosheets had high conductivity without reduced transparency. Therefore, this UV irradiation method, modified by adding metal ions, is quite effective at improving and diversifying properties of Au nanosheets.

## 1. Introduction

Two-dimensional (2D) conductive materials, such as metal–organic framework thin films and crystalline conjugated polymer films, have attracted extensive research attention because of their significant potential for broad opto-electronic applications [1,2,3,4,5,6]. In order to fully realize the properties of these materials, it is important to establish a fabrication method that controls the composition, structure, and even the size of 2D materials. Various fabrication techniques based on top-down and bottom-up strategies have been proposed for 2D conductive materials [3,4]. In bottom-up strategies, the use of interfaces as a fabrication site for 2D materials has received considerable attention, because interfaces, especially gas–liquid interfaces, can serve as molecularly flat and defect-free substrates [1]. Large-area and free-standing metal–organic framework films and conjugated polymer films have been successfully synthesized using the air–water interface [1,2,3,4,5,6].

For advanced opto-electronic devices, transparent conductive materials are also essential components [7,8,9,10,11,12,13,14,15,16]. Although the most commonly used transparent conducting films are fluorine- or indium-doped tin oxide, they have low flexibility and are brittle [17,18,19,20]. Noble metal nanoparticles (NPs) have attracted attention as alternative materials. For example, flexible transparent conductive films have been fabricated by arranging NPs in grid-like patterns on appropriate substrates [9,10,21,22,23,24]. However, post-processing surfaces of NPs is essential to removing insulating organic compounds to make the grids conductive [10,23,24,25,26]. Despite the requirement for post-processing, this method remains appealing for fabricating transparent conductive films due to the availability of various synthesis protocols for the desired sizes of metal NPs.

In previous papers, we have demonstrated that UV irradiation of non-conductive AuNP monolayers at the air–water interface can be converted into flexible, transparent, conductive Au nanosheets [27,28]. In this process, the key to conversion by UV irradiation is photodecomposition of capping molecules covering AuNPs to form an unstable, bare Au surface, and aggregation and fusion phenomena at those surfaces. Here, fusion of AuNPs depends on their size. Small particles fuse easily, but the fusion process is less likely to proceed with larger particles. Actually, in a previous study, we demonstrated that UV irradiation of monolayers of AuNPs 30 nm in diameter created connections among them, but they did not fuse sufficiently, resulting in nanosheets that retained the shapes of the particles [28]. On the other hand, AuNPs with diameters of 15 nm or less gave rise to flat nanosheets of uniform thickness, indicating that fusion between AuNPs was sufficient. Accordingly, this UV irradiation technique appears to be inapplicable for larger AuNPs. Nanosheets made from these larger particles exhibit high mechanical strength and conductivity, although their conductivity and mechanical strength are inversely related to their transparency.

Furthermore, to change the composition of a nanosheet, it is necessary to synthesize NPs with the desired composition. An alternative is to synthesize two NPs with different metals and arrange them uniformly on water. Both methods, however, require considerable effort. Hence, to further develop this UV irradiation technique, a strategy is needed to produce nanosheets with various thicknesses and compositions from a single type of metal nanoparticle monolayer. In this study, we propose a UV irradiation technique for preparing thick Au nanosheets from a single type of AuNP by replacing the water subphase with aqueous solutions of HAuCl_4_. In addition, we investigated whether the composition of Au nanosheets can be controlled by changing the metal species in solution. Here, silver, which alloys with gold and has a smaller ionization tendency than gold, was selected. The silver electrolytes used were AgNO_3_, an inorganic ion, and CH_3_COOAg, an organic ion, as an anion. We demonstrate that using AgNO_3_ and CH_3_COOAg gives rise to Au-Ag nanosheets with high conductivity and transparency and we show that their morphologies depend on the electrolytes chosen. Au-Ag nanosheets prepared from AgNO_3_ have homogeneous flat structures, while those from CH_3_COOAg have the structure of AuNPs embedded in a Ag thin film.

## 2. Materials and Methods

### 2.1. Materials

Silver acetate (CH_3_COOAg), silver nitrate (AgNO_3_), sodium borohydride (NaBH_4_), and chloroform were obtained from Kanto Chemical Co., Inc. (Tokyo, Japan) Gold (III) chloride trihydrate (HAuCl_4_·3H_2_O), tetraoctylammonium bromide (TOAB), and dodecanethiol (DDT) were purchased from Nacalai Tesque Inc. (Kyoto, Japan), Tokyo Chemical Industry Co., Ltd. (Tokyo, Japan), and Sigma-Aldrich (St. Louis, MO, USA), respectively. All chemicals were used as received, without further purification.

### 2.2. Preparation of Au Nanoparticles (AuNPs)

AuNPs capped with DDT were prepared according to a modified Brust–Schiffrin method [29]. A 0.384 mmol HAuCl_4_ aqueous solution (7.91 mL) was added to a 0.362 mmol phase-transfer reagent, TOAB, dissolved in toluene (7.91 mL), and the mixture was stirred for 15 min or until the aqueous phase was clear. The toluene phase, containing phase-transferred gold, was subsequently collected and a 0.066 mmol DDT solution in toluene (1.51 mL) was added. After the solution was stirred for 1 h, a freshly prepared 4.22 mM solution of NaBH_4_ in water (2 mL) was quickly added with high stirring to reduce the Au ions. The reaction mixture was vigorously stirred for 12 h to allow formation of AuNPs and the organic phase was isolated. As-prepared AuNPs were then dispersed in DDT (2.54 mL), and the dispersion was refluxed at 80 °C for 22 h. In order to remove excess DDT, AuNPs in DDT were washed thoroughly several times with acetone through centrifugal separation. Purified AuNPs were dispersed in chloroform. AuNPs had a spherical shape and an average diameter of 4.7 ± 0.5 nm.

### 2.3. Preparation of Au and Au-Ag Nanosheets

A suspension of purified AuNPs in chloroform was spread on water and compressed using a Teflon barrier until AuNPs were organized into a close-packed structure. The AuNP monolayer was moved onto an aqueous solution of HAuCl_4_, AgNO_3_, or CH_3_COOAg, and the AuNP monolayer was irradiated with UV light through an optical fiber (Figure 1). Here, concentrations of 4.8, 48, and 480 µM were used for aqueous solutions of HAuCl_4_ and 5, 50, and 500 µM for aqueous solutions of AgNO_3_ and CH_3_COOAg. Au nanosheets were prepared by UV irradiation of a close-packed AuNP monolayer prepared on water without transferring it to another aqueous solution. UV light from a 250 W mercury lamp (REX-250, Asahi Spectra Co., Ltd., Tokyo, Japan) was passed through a narrow band optical filter to obtain 248 nm UV light. Samples were irradiated with 248 nm UV light at an intensity of ≈60 mW cm^−2^.

### 2.4. Characterization

To characterize Au nanosheets, as-prepared Au or Au-Ag nanosheets on aqueous solutions were transferred from the water surface to various solid substrates using the surface-lowering method: (i) copper grids coated with an elastic carbon film for transmission electron microscopy (TEM); (ii) glass slides for scanning electron microscopy (SEM), X-ray diffraction measurements (XRD), and electrical resistivity measurements; (iii) quartz plates for UV-vis analysis; and (iv) quartz crystal chips for quartz crystal microbalance (QCM) measurements.

TEM images were taken using a JEOL JEM-1101 microscope (Tokyo, Japan) operated at 100 kV. High-resolution (HR) TEM images, energy-dispersive X-ray spectroscopy (EDS) mapping images and scanning TEM images were captured using a JEOL JEM-2100 microscope operated at 200 kV. SEM images were obtained on a Carl Zeiss SUPRA40 microscope operating at 8 kV. UV-visible spectra were acquired on a JASCO V-780 UV-vis spectrophotometer. X-ray diffraction (XRD) measurements were performed using a diffractometer (Rigaku, Ultima IV, Tokyo, Japan). Electrical resistivity of nanosheets deposited on the glass substrates was measured using a four-probe method with a digital multimeter (NF Corporation, DM2561A, Kanagawa, Japan) and four-point probe (Astellatech, Inc., Kanagawa, Japan), where the spacing between adjacent probes was 1 mm.

Masses of Au and Au-Ag nanosheets transferred onto 9 MHz AT-cut chips (SEIKO EG&G, QA-A9M-AU, Tokyo, Japan) were monitored using the QCM SYSTEM (SEIKO EG&G, QCM922). Masses of nanosheets were evaluated using the following Sauerbrey equation [30,31]:$$∆F=-\frac{2∆mf_{0}^{2}}{A\sqrt{\mu_{q}\rho_{q}}}$$
where ∆F is the measured frequency change (Hz), $f_{0}$ is the fundamental frequency, ∆m is the mass change (g), A is the electrode area (cm^2^), $\rho_{q}$ is the density of quartz (2.65 g cm^−3^), and $\mu_{q}$ is the shear modulus of quartz (2.95 × 10^11^ g cm^−1^ s^−2^).

## 3. Results

### 3.1. Preparation of Thick Au Nanosheets

AuNP monolayers were prepared by spreading a chloroform dispersion of DDT-capped AuNPs (4.7 ± 0.5 nm) on water, and the monolayer was irradiated with UV light for 60 min. AuNPs in the monolayer were transformed into a mesh-like Au nanosheet (Figure 2a), as reported previously [27]. To investigate effects of metal species in water on formation of Au nanosheets, after preparing an AuNP monolayer on water, the monolayer was moved onto an aqueous HAuCl_4_ solution and then the monolayer was irradiated with UV light. Figure 2b–d show TEM images of resultant Au nanosheets prepared on 4.8, 48, and 480 µM aqueous HAuCl_4_ solutions. All AuNP monolayers were transformed into mesh-like Au nanosheets upon UV irradiation. However, the mesh size varied significantly with the concentration of HAuCl_4_. The width of the mesh prepared on water was 9.5 ± 2.8 nm, whereas those prepared on 48 and 480 µM HAuCl_4_ solutions were 25.8 ± 7.8 and 81.9 ± 29.5 nm, respectively. This increase in mesh size was also apparent in SEM images (Figure 3).

TEM and SEM images (Figure 2 and Figure 3) indicate that the amount of Au in these nanosheets increases with increasing concentrations of HAuCl_4_. The mass of each Au nanosheet was then measured using the QCM technique to evaluate how much the mass increased. The evaluated masses for water and 48 and 480 µM HAuCl_4_ samples were 1.33, 4.09, and 13.8 µg/cm^2^, respectively. Based on these masses, the corresponding nanosheet thicknesses were calculated as 3.5, 10.8, and 36.5 nm, assuming a uniform gold film and gold density of 19.32 g/cm^3^. The increase in mass indicates that beside AuNPs on water, Au atoms produced by the reduction of AuCl_4_^−^ ions were the source of the Au nanosheet. Accordingly, Au nanosheets on HAuCl_4_ solutions were formed by fusion of AuNPs and deposition of Au atoms. Here, reduction of AuCl_4_^−^ ions may occur near the AuNP monolayer, because the color of HAuCl_4_ solutions did not change after UV irradiation, remaining the original pale yellow. Further, UV irradiation of aqueous HAuCl_4_ solutions for an additional 60 min without an AuNP monolayer caused no change in the solution, meaning that AuCl_4_^−^ ions are not reduced by UV irradiation alone.

Figure 4 shows the TEM images of UV-irradiated AuNP monolayers on water and aqueous HAuCl_4_ solutions as a function of time. After 15 min of UV irradiation, AuNPs on water remained almost unchanged, but those on HAuCl_4_ solutions transformed completely into larger aggregates. In particular, on the 480 µM HAuCl_4_ solution, the morphological change of AuNPs was remarkable. Hence, HAuCl_4_ promotes the morphological transformation of AuNPs to nanosheets.

To characterize Au nanosheets, XRD, electrical resistance, and transmittance in the visible region were measured. Figure 5 shows the XRD pattern of a nanosheet prepared on a 480 µM HAuCl_4_ solution. The nanosheet was confirmed as crystalline gold, based on reflection peaks at 38.2°, 44.3°, 64.5°, 77.5°, and 81.7°, which were assigned to the (111), (200), (220), (311), and (222) crystal facets of Au, respectively, with a face-centered cubic structure [32,33]. Electrical resistances of Au nanosheets prepared on water, 4.8 µM, 48 µM, and 480 µM HAuCl_4_ solutions were 2600 Ω/sq, 2220 Ω/sq, 6.5 Ω/sq, and 4.1 Ω/sq, respectively. Electrical resistances of Au nanosheets prepared on HAuCl_4_ solutions were considerably smaller, which can be explained by the increase in film thickness, as evaluated by QCM measurements. Hence, we demonstrated that Au nanosheets with high electrical conductivity can be prepared using a single type of AuNP, avoiding the necessity of synthesizing new AuNPs with different diameters.

Electrical resistance and transparency are inversely related, and both properties depend strongly on the thickness of Au nanosheets. Thick Au nanosheets possess high electrical conductivity but low transparency, and vice versa [34,35]. Thus, as a natural consequence, Au nanosheets prepared on HAuCl_4_ solutions are likely to have low transparency in the visible and UV region. Actually, Au nanosheets prepared on 48 µM and 480 µM HAuCl_4_ solutions had optical transparencies of 55% and 8%, respectively (Figure 6), whereas the Au nanosheet prepared on water had a transparency of ~80%.

### 3.2. Preparation of Au-Ag Hybrid Nanosheets

#### 3.2.1. Effect of AgNO_3_ and CH_3_COOAg

Employing HACl_4_ solutions as a subphase was useful to modify Au nanosheets prepared by irradiating AuNP monolayers with UV light. This suggested that other metal ions might also be useful for modifying Au nanosheets. We then attempted to prepare Au-Ag nanosheets by irradiating AuNP monolayers on aqueous AgNO_3_ and CH_3_COOAg solutions, using the same method as described above. Figure 7 shows TEM images of UV-irradiated AuNPs on water, aqueous 500 µM AgNO_3_, and CH_3_COOAg solutions. For water and AgNO_3_ systems (Figure 7a,b,d,e), UV irradiation resulted in complete fusion of AuNPs, yielding flat nanosheets with gaps in which no metal was present, although the gap size and density of the AgNO_3_ nanosheets were smaller than those on water.

Reducing the size and density may be attributed to depositing Ag atoms generated by photoredution of Ag^+^ ions. Deposition of Ag atoms was confirmed by TEM-EDS (energy-dispersive X-ray spectroscopy) mapping. Figure 8a–c show that a nanosheet prepared on aqueous 500 µM AgNO_3_ is composed of both Ag and Au, and these metals are homogeneously distributed throughout the nanosheet. The molar ratio of Ag to Au in the nanosheet was Ag/Au = 20/80 from TEM-EDS analysis. Further, we measured the XRD pattern of the hybrid nanosheet to determine whether silver is oxidized or whether AgNO_3_ deposits directly on the nanosheet, although distinguishing between metallic Au and Ag from XRD diffraction is difficult because these metals have nearly identical lattice constants [36]. The XRD pattern in Figure 9a showed reflection peaks at 38.2°, 44.0°, 64.3°, and 77.3°, assigned to the (111), (200), (220), and (311) crystal facets of metallic Ag and/or Au, respectively [32,33,37,38]. The absence of silver oxide and AgNO_3_ peaks suggested that the hybrid nanosheet was composed of metallic Ag and Au [39,40]. Accordingly, the nanosheet prepared on AgNO_3_ aqueous solutions is formed by Ag atoms supplied from the aqueous phase, as well as the fusion of AuNPs, leading to fewer and smaller gaps.

With regard to the CH_3_COOAg system in Figure 7c,f, the nanosheet morphology differed significantly from that of the AgNO_3_ system. Outlines of AuNPs were clearly visible in TEM images of the nanosheet, and AuNPs still remained even after UV irradiation for 60 min. There was no product between AuNPs before UV irradiation (Figure 7i). However, observing TEM images of AuNPs after UV irradiation (Figure 7f), we noticed that AuNPs were connected by a thin film. TEM-EDS mapping (Figure 8b) showed that elemental Ag is present throughout the nanosheet, and TEM-EDS analysis estimated the composition of the hybrid nanosheet as Ag/Au = 40/60. Further, the XRD pattern of the hybrid nanosheets (Figure 9b) showed no peaks for silver oxide or CH_3_COOAg, and all peaks were assigned to metallic Ag and/or Au. Hence, these TEM-EDS observations and the XRD analysis suggest that metallic silver deposits as a thin film on the entire AuNP monolayer. This assessment is also consistent with the fact that the average diameter of AuNPs remained almost remained unchanged after UV irradiation (Figure 7g,h).

#### 3.2.2. Effect of CH_3_COOAg and AgNO_3_ Concentrations

We demonstrated that UV irradiation of AuNP monolayers on a CH_3_COOAg aqueous solution yields a hybrid nanosheet consisting of an AuNP monolayer covered with a Ag thin film, whereas a flat Au nanosheet was obtained on water without CH_3_COOAg. Thus, the morphology of these nanosheets is expected to be highly dependent on the CH_3_COOAg concentration. The appearance of nanosheets prepared on 5 µM CH_3_COOAg was uniform and similar to that prepared on water, (Figure 10a,d). On the other hand, the morphology at 50 µM CH_3_COOAg (Figure 10b,e) was not uniform and consisted of two domains. One domain comprised fused AuNPs (red circle in Figure 10e) and the other comprised AuNPs covered with a Ag thin film (blue circle), similar to that of the 500 µM CH_3_COOAg nanosheet (Figure 10c,f). The content of Ag in the hybrid nanosheets was Ag/Au = 19/81 for 5 µM, 35/65 for 50 µM and 40/60 for 500 µM. At low concentrations of CH_3_COOAg, formation of uniform nanosheets is probably attributable to preferential fusion between AuNPs, rather than to supplying Ag atoms generated by photoreduction of Ag ions, as shown in Figure 11a. On the contrary, at high concentrations (Figure 10b), since many photoreduced Ag atoms are supplied, prior to fusion of AuNPs, Ag atoms preferentially deposit on AuNPs, which probably inhibits AuNP fusion. Thus, the domain structure at the middle concentration of 50 µM is thought to be driven by competition between Ag deposition on AuNPs and fusion of AuNPs.

On the other hand, the morphology of nanosheets prepared on 500 µM AgNO_3_ was almost identical to that prepared on water (Figure 7). Further, the appearance of nanosheets prepared on 4.8 µM and 48 µM AgNO_3_ was also uniform and similar to that prepared on water, indicating that the morphology in the AgNO_3_ system was independent of the concentration of AgNO_3_. We demonstrated that Au-Ag hybrid nanosheets can be prepared on aqueous solutions of AgNO_3_ and CH_3_COOAg, but their morphology differed significantly. As the Ag/Au ratio was 40/60 for the 500-µM CH_3_COOAg system and 20/80 for the 500 µM AgNO_3_ system, this silver content difference probably causes the morphology difference. This is because larger amounts of Ag atoms can cover AuNPs, leading to AuNPs embedded in a Ag film. Why is the supply of Ag atoms in the CH_3_COOAg system greater than in the AgNO_3_ system? According to previous reports [41,42], UV irradiation of aqueous organic molecules, such as alcohols and carboxylic acids, promotes photoreduction of metal ions, because these organic molecules photo-generate radical species that have the potential to reduce metal ions, including Ag. Hence, the larger quantity of reduced Ag in the CH_3_COOAg system is because CH_3_COO^−^ ions generate radical species that can reduce Ag ions. Thus, more silver ions are reduced in the CH_3_COOAg system than in the AgNO_3_ system.

### 3.3. Conductivity and Transparency of Au-Ag Hybrid Nanosheets

The electrical resistance and optical transmittance of Au-Ag nanosheets were evaluated. Electrical resistances of nanosheets prepared on 500 µM AgNO_3_ and CH_3_COOAg solutions were 1600 Ω/sq and 83 Ω/sq, respectively, smaller than that of the original Au nanosheet (2600 Ω/sq). In particular, the nanosheet prepared on 500 µM CH_3_COOAg solution, which has a Ag thin film covering the AuNP monolayer (Figure 7f), had low electrical resistance. On the other hand, transmittances in the UV and visible regions of Au-Ag nanosheets (Figure 12) were almost identical to that of the original Au nanosheet. Hence, covering AuNP monolayers with a Ag thin film improved the conductivity considerably without degrading the transparency. Accordingly, in the process of fabricating transparent conductive films with AuNP monolayers, deposition and coating with silver under UV irradiation proved to be remarkably useful in improving their properties.

## 4. Conclusions

We demonstrated that the versatility of a UV irradiation technique for preparing metal nanosheets from metal NP monolayers can be enhanced using subphases containing metal ions. With a single type of AuNP monolayer, using aqueous HAuCl_4_, AgNO_3_, and CH_3_COOAg solutions as alternative subphases, morphology, and composition of Au nanosheets can be changed. Without synthesizing AuNPs of different diameters, one type of AuNP can be used to prepare Au nanosheets with high conductivity but low transparency using aqueous HAuCl_4_ solutions. Using aqueous AgNO_3_ and CH_3_COOAg solutions, Au-Ag nanosheets with high conductivity and transparency can be prepared from the same AuNPs. Further, Au-Ag nanosheets prepared from AgNO_3_ have a homogeneous flat structure, while those from CH_3_COOAg have the structure of AuNPs embedded in a Ag thin film.

## Figures and Tables

**Figure 1 materials-17-00842-f001:**
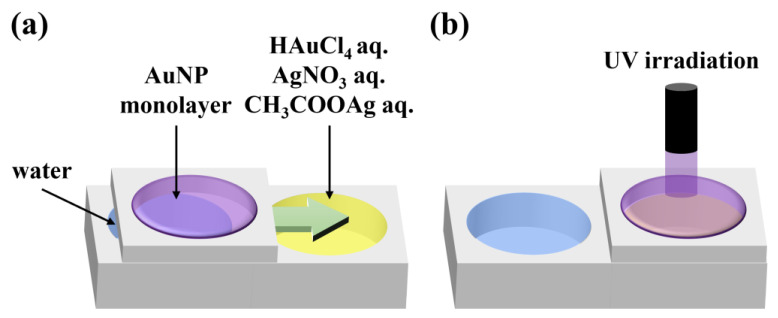
Schematic illustration of the experimental procedure. (**a**) AuNP monolayers are prepared by spreading a chloroform dispersion of AuNPs on water. The AuNP monolayer is then transferred onto an aqueous solution of HAuCl_4_, AgNO_3_, or CH_3_COOAg, and (**b**) irradiated with UV light.

**Figure 2 materials-17-00842-f002:**
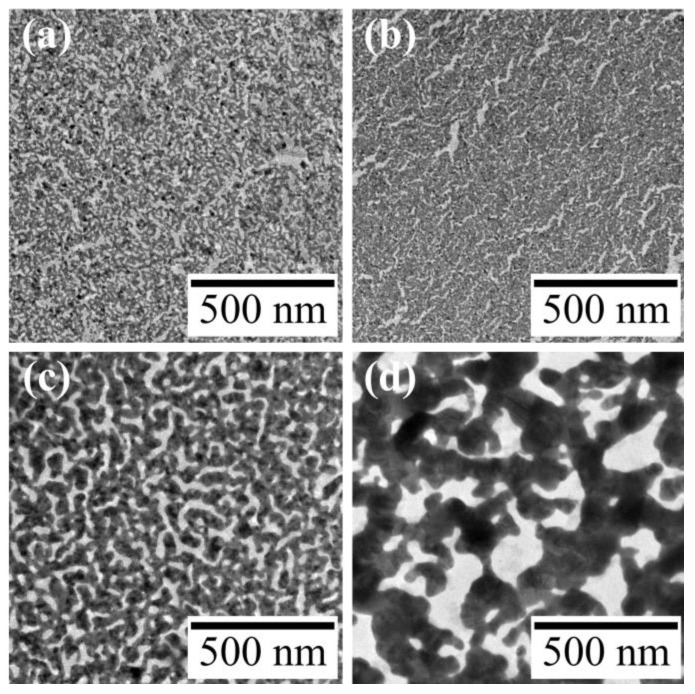
TEM images of Au nanosheets prepared on (**a**) water, (**b**) 4.8-, (**c**) 48-, and (**d**) 480 μM HAuCl_4_ aqueous solutions.

**Figure 3 materials-17-00842-f003:**
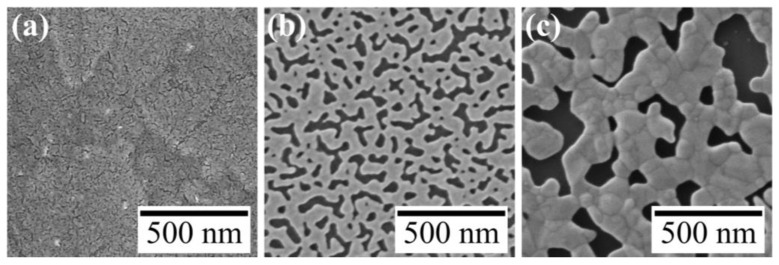
SEM images of Au nanosheets prepared on (**a**) 4.8-, (**b**) 48-, and (**c**) 480 μM HAuCl_4_ aqueous solutions.

**Figure 4 materials-17-00842-f004:**
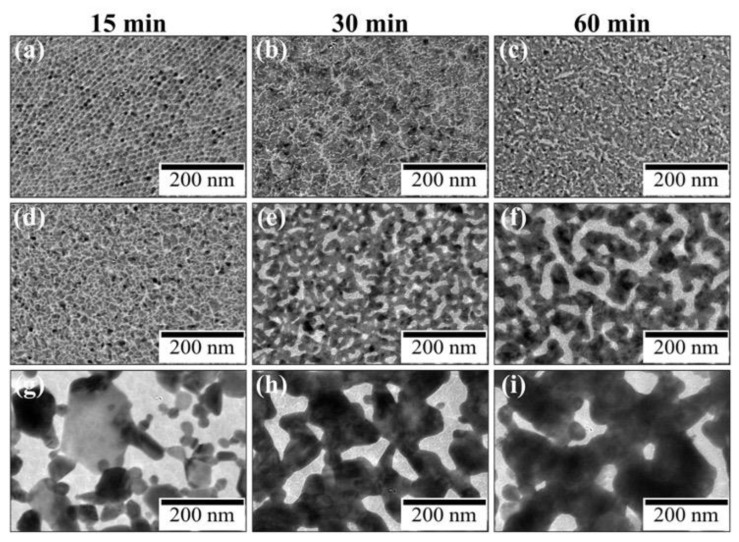
TEM images of UV-irradiated AuNP monolayers on (**a**–**c**) water, (**d**–**f**) 48-, and (**g**–**i**) 480 μM HAuCl_4_ aqueous solutions at irradiation times of (**a**,**d**,**g**) 15, (**b**,**e**,**h**) 30, and (**c**,**f**,**i**) 60 min.

**Figure 5 materials-17-00842-f005:**
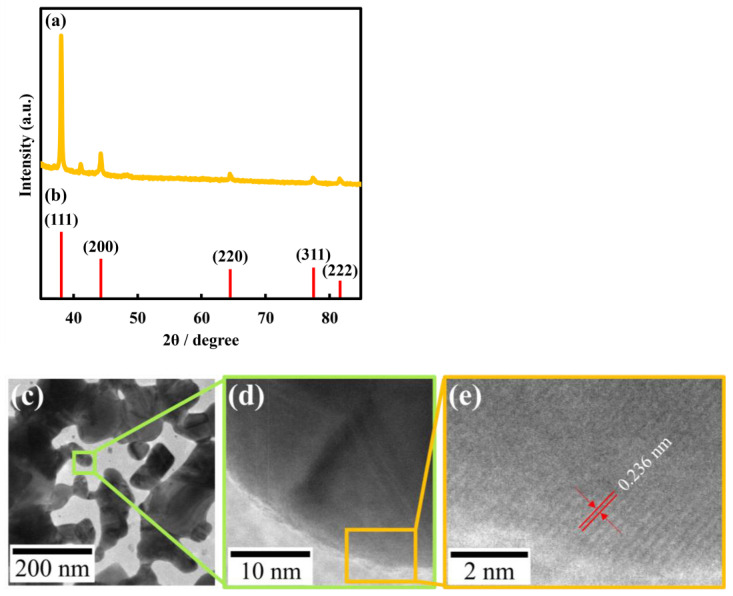
XRD patterns of (**a**) Au nanosheets prepared on a 480 µM HAuCl_4_ aqueous solution and (**b**) Au (JCPDS file: 04-0784). (**c**) TEM image and (**d**,**e**) HR-TEM images of Au nanosheets prepared on a 480 µM HAuCl_4_ aqueous solution. (**e**) Periodical fringe of 0.236 nm corresponds to Au (111) crystal facet.

**Figure 6 materials-17-00842-f006:**
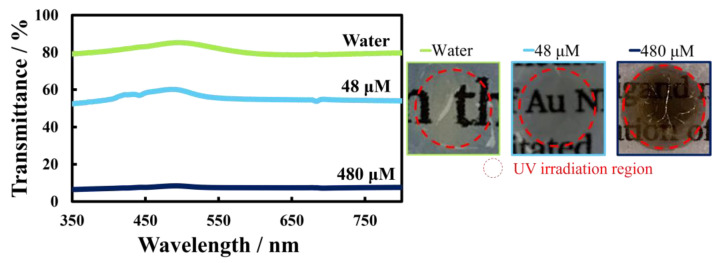
UV-vis spectra of Au nanosheets prepared at various concentrations of HAuCl_4_ aqueous solution, and photographs of Au nanosheets deposited on quartz plates.

**Figure 7 materials-17-00842-f007:**
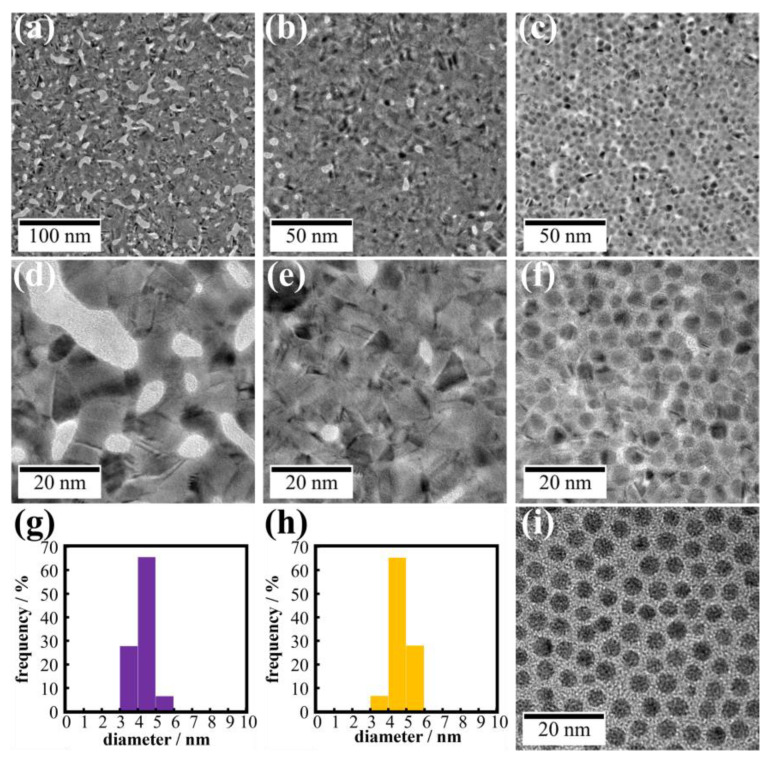
TEM images of Au nanosheets prepared on (**a**,**d**) water, (**b**,**e**) AgNO_3_, and (**c**,**f**) CH_3_COOAg aqueous solutions. Particle size distribution histograms of (**g**) AuNPs before UV irradiation and (**h**) AuNPs in a nanosheet prepared on CH_3_COOAg aqueous solution. (**i**) TEM image of AuNPs before UV irradiation.

**Figure 8 materials-17-00842-f008:**
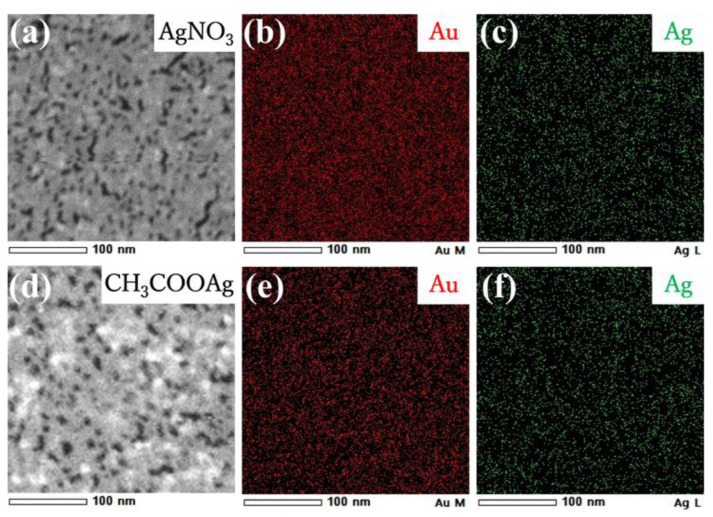
STEM images (**a**,**d**) and EDS elemental mapping images (**b**,**c**,**e**,**f**) of Au-Ag nanosheets prepared on (**a**–**c**) AgNO_3_ and (**d**–**f**) CH_3_COOAg aqueous solutions.

**Figure 9 materials-17-00842-f009:**
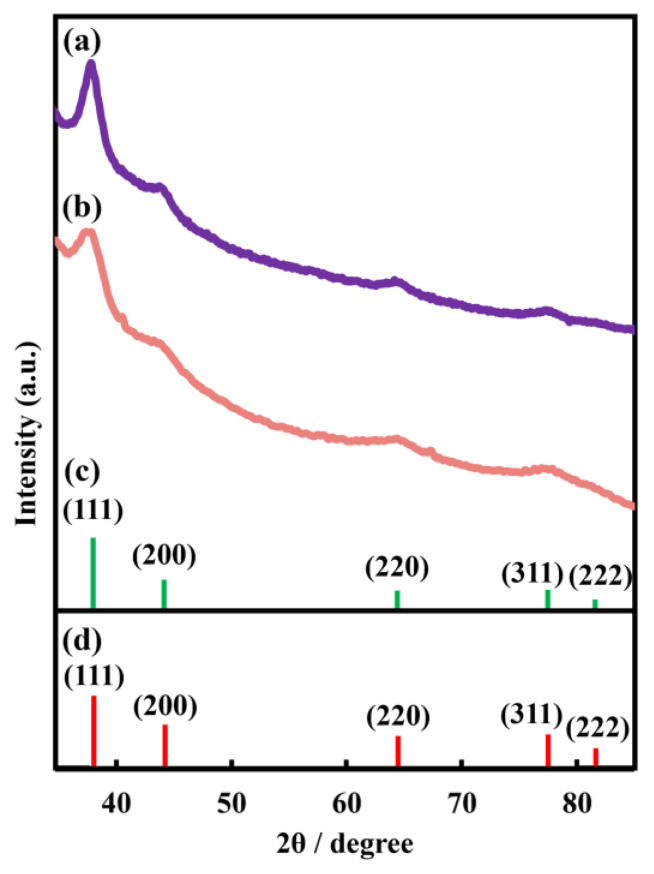
XRD patterns of Au-Ag nanosheets prepared on (**a**) AgNO_3_ and (**b**) CH_3_COOAg aqueous solutions, (**c**) Ag (JCPDS file: 04-0783), and (**d**) Au (JCPDS file: 04-0784).

**Figure 10 materials-17-00842-f010:**
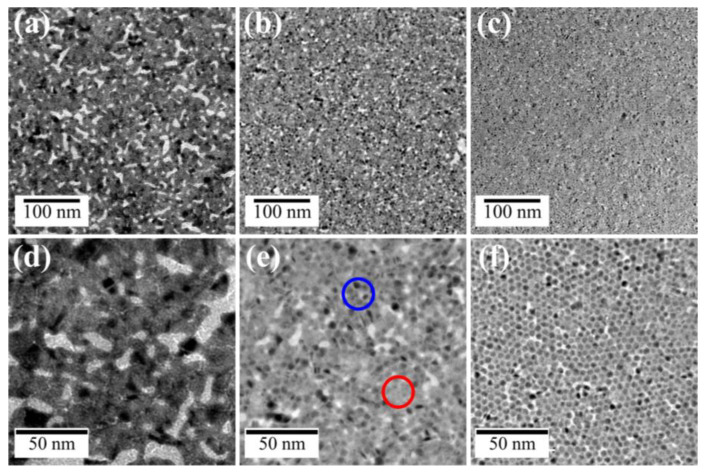
TEM images of Au-Ag nanosheets prepared on (**a**,**d**) 5-, (**b**,**e**) 50-, and (**c**,**f**) 500 μM CH_3_COOAg aqueous solutions. Blue circle: domain of AuNPs covered with a Ag thin film, red circle: fused domain of AuNPs.

**Figure 11 materials-17-00842-f011:**
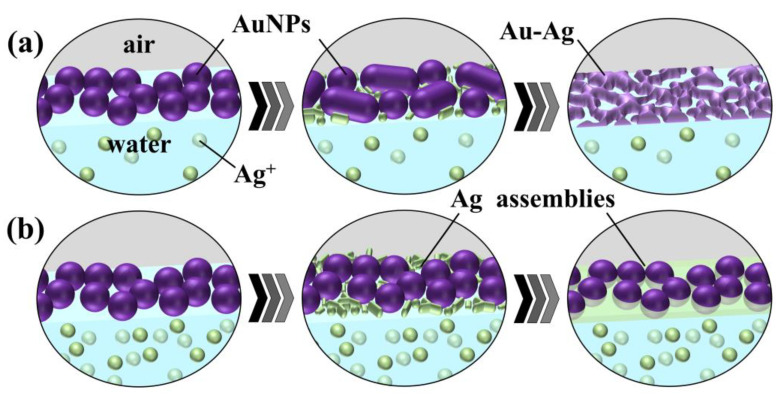
Schematic illustration of proposed growth mechanism of Au-Ag hybrid nanosheets at (**a**) a low concentration and (**b**) a high concentration of CH_3_COOAg aqueous solutions.

**Figure 12 materials-17-00842-f012:**
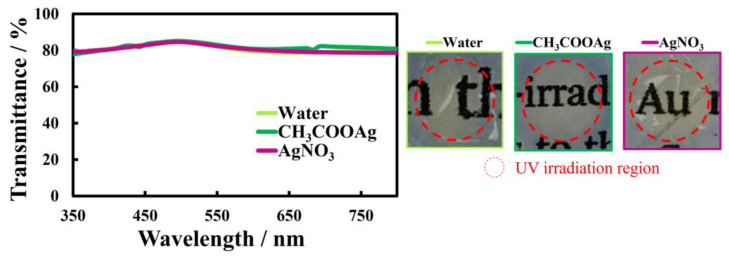
UV-vis spectra of Au and Au-Ag nanosheets prepared on AgNO_3_ and CH_3_COOAg aqueous solutions, and photographs of Au and Au-Ag nanosheets deposited on quartz plates.

## Data Availability

Data presented in this study are available on request from the corresponding author, but they are not publicly available because they pertain to an on-going project.

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
