# Peer review of "Conductive Nanosheets Fabricated from Au Nanoparticles on Aqueous Metal Solutions under UV Irradiation"

_materials, 2024, doi:10.3390/ma17040842_

Round 1

Reviewer 1 Report

Comments and Suggestions for Authors

This manuscript entitled “Conductive nanosheets fabricated from Au nanoparticles on aqueous metal solutions under UV irradiationexhibit highly conductive nanosheets prepared by simple UV iddadiation.. The paper has been well-prepared and the length is suitable, thus it is suggested to publish this manuscript in a minor revision:

1.     The standard XRD PDF card of Au is suggested to be included in the XRD pattern. Also, high-resolution TEM is suggested to be included to further support the crystallinity in this work.

2.     The as-prepared metal thin film is conductive and can be potentially applied in the electronic devices. Therefore, it is important to prepare self-support thin film to realize this application. How about the self-support performance of this thin film? Please provide some discussions.

3.     Ag nanoparticles can effectively improve the transmittance of the thin film, as shown in figure 12. Please explain the underlying mechanism.

4.     In the mechanism of Au-Ag nanosheet assemblies, what is the driving force to make the Au and Ag particles to orderly fabricate? What role the air-solution interface play in this assembly process? Please provide some discussions.

5.      Some important progress towards air-solution assembly and optoelectronic semiconductors are suggested to be included in the introduction part, such as  https://doi.org/10.1016/j.pnsc.2022.03.006, J. Phys. Chem. Lett. 13 (2022) 1215-1225

Comments on the Quality of English Language

No suggestions

Author Response

Thank you for your review.

Reviewer: 1

Comment 1) The standard XRD PDF card of Au is suggested to be included in the XRD pattern. Also, high-resolution TEM is suggested to be included to further support the crystallinity in this work.
Answer: Thank you for your advice. In Figure 5 and Figure 9, XRD card data of Au and Ag were added. Further, In Figure 5, high-resolution TEM images of Au nanosheets were added.

Comment 2) The as-prepared metal thin film is conductive and can be potentially applied in the electronic devices. Therefore, it is important to prepare self-support thin film to realize this application. How about the self-support performance of this thin film? Please provide some discussions.
Answer: Thank you for your comment. Sorry, we did not study the self-supporting performance of Au and Au-Ag nanosheets in detail. The performance depends on both the thickness and area of the nanosheets. So far, we know only that all nanosheets are stable as freestanding in a 30 µm circle frame (TEM mesh without carbon or polymer support). And in case of thick Au nanosheets (Fig. 3c), they were stable as freestanding film in a 1 mm square frame of metal wire, but thin nanosheets were not freestanding (they were torn during the drying process).

Comment 3) Ag nanoparticles can effectively improve the transmittance of the thin film, as shown in figure 12. Please explain the underlying mechanism.
Answer: We don’t believe that deposited Ag can effectively improve the transmittance of the original Au nanosheets, only that deposited Ag do not reduce the original transmittance. Namely, since the thickness of the deposited Ag film (not Ag NPs) was thin, transmittance of light was practically unchanged. The thicknesses of Au and Au-Ag nanosheets estimated from QCM measurements were 3.4 nm (Au), 4.2 nm (Au-Ag), respectively.

Comment 4) In the mechanism of Au-Ag nanosheet assemblies, what is the driving force to make the Au and Ag particles to orderly fabricate? What role the air-solution interface play in this assembly process? Please provide some discussions.
Answer: In Au-Ag nanosheets, silver is present as a thin film, not as nanoparticles.
Before UV irradiation, we prepared hexagonal packed AuNP monolayer on water (or on solutions) using Langmuir technique (as described in section 2.3 preparation of Au and Au-Ag nanosheets). Ag atoms produced by UV irradiation are deposited onto AuNP monolayer from the solution side, and the deposited Ag atoms form a thin film (not NPs) under the AuNPs monolayer. Accordingly, the ordered array structure of AuNPs in Au-Ag nanosheets is derived from maintaining the initial arrangement of AuNPs.

Comment 5) Some important progress towards air-solution assembly and optoelectronic semiconductors are suggested to be included in the introduction part, such as  https://doi.org/10.1016/j.pnsc.2022.03.006, J. Phys. Chem. Lett. 13 (2022) 1215-1225
Answer: In Introduction, the following sentences and References [1]–[6] were added.

“Two-dimensional (2D) conductive materials, such as metal-organic framework thin films and crystalline conjugated polymer films, have attracted extensively research attention because of their significant potential for broad opto-electronic applications [1–6]. In order to fully realize the properties of these materials, it is important to establish a fabrication method that controls the composition, structure, and even the size of 2D materials. Various fabrication techniques based-on top-down and bottom-up strategies have been proposed for 2D conductive materials [3, 4]. In the bottom-up strategy, the use of interfaces as a fabrication site for 2D materials has received considerable attention, because the interface, especially the gas-liquid interface, can serve as a molecularly flat and defect-free substrate [1]. Large-area and free-standing metal-organic framework films and conjugated polymer films have been successfully synthesized the air-water interface [1–6].”

Reviewer 2 Report

Comments and Suggestions for Authors

The authors have reported on conductive nanosheets made using Au nanoparticles on Au or Ag solutions. While this work is novel and interesting it needs a major rewrite mainly of the abstract, introduction and experimental part. The abstract is too general and does not give the motivation for the work nor major results. It is not clear why the authors used Ag besides Au. In the last paragraph of the Introduction - first sentence the authors mention compositions of a single type of nanoparticles, yet here they used Au and Ag.

What was the motivation for using Ag? Why two different Ag solutions - nitrate and acetate? The authors should clearly explain and describe the motivation for this work, what was obtained and how this is of interest.

In the results we can see that the authors varied the solution concentrations, yet this is not described at all in the experimental part. This should be changed and the experimental part needs to contain all experimental data.

The solution concentrations for Au were 4.8, 48 and 480 and for silver acetate 5, 50 and 500, why not the same? Why did the authors choose these particular concentrations? Why not vary the silver nitrate concentration too?

Why did the mesh size vary for Au? This has not been explained clearly. Didi it vary for Au also?

Author Response

Thank you for your review.

Comment 1) The authors have reported on conductive nanosheets made using Au nanoparticles on Au or Ag solutions. While this work is novel and interesting it needs a major rewrite mainly of the abstract, introduction and experimental part. A) The abstract is too general and does not give the motivation for the work nor major results. It is not clear why the authors used Ag besides Au. B) In the last paragraph of the Introduction - first sentence the authors mention compositions of a single type of nanoparticles, yet here they used Au and Ag.

Answer: Thank you for your comments.

A) Abstract was changed as follows:

Original: Highly transparent, conductive nanosheets are extremely attractive for advanced optoelectronic applications. Previously, we have demonstrated that transparent, conductive Au nanosheets can be prepared by UV irradiation of Au nanoparticle (AuNP) monolayers spread on water, which serves as the subphase. In this study, we show that this UV irradiation technique can easily be extended as a nanosheet modification method by using subphases containing metal ions. That is, aqueous solutions containing Au and Ag ions are effective in modifying Au nanosheets. HAuCl4 solutions increased Au nanosheet thickness, which improved conductivity, but reduced transparency. On the other hand, use of aqueous AgNO3 and CH3COOAg solutions yielded Au-Ag hybrid nanosheets; however, their morphologies depended on the electrolytes used. Homogeneous Au-Ag nanosheets from AgNO3 and AuNPs covered with an Ag nanosheet from CH3COOAg were also produced. Further, these Au-Ag hybrid nanosheets had high conductivity without reduced transparency. Therefore, this UV irradiation method, modified by adding metal ions is quite effective at improving and diversifying properties of Au nanosheets.

Revised: Highly transparent, conductive nanosheets are extremely attractive for advanced optoelectronic applications. Previously, we have demonstrated that transparent, conductive Au nanosheets can be prepared by UV irradiation of Au nanoparticle (AuNP) monolayers spread on water, which serves as the subphase. However, thick Au nanosheets cannot be fabricated because the method is not applicable to large Au NPs. Further, in order to fabricate nanosheets with different thicknesses and compositions, it is necessary to prepare the appropriate NPs. A strategy is needed to produce nanosheets with different thicknesses and compositions from a single type of metal NP monolayer. In this study, we show that this UV irradiation technique can easily be extended as a nanosheet modification method by using subphases containing metal ions. That is, aqueous solutions containing Au and Ag ions are effective in modifying Au nanosheets. UV irradiation of 4.7 nm AuNP monolayers on 480 µM HAuCl4 solution increased the thickness of Au nanosheets from 3.5 nm to 36.5 nm, which improved conductivity, but reduced transparency. On the other hand, use of aqueous AgNO3 and CH3COOAg solutions yielded Au-Ag hybrid nanosheets; however, their morphologies depended on the electrolytes used. In Au-Ag nanosheets prepared on aqueous 500 µM AgNO3, Au and Ag metals are homogeneously distributed throughout the nanosheet. On the other hand, in Au-Ag nanosheets prepared on aqueous 500 µM CH3COOAg, AuNPs still remained and these AuNPs were covered with an Ag nanosheet. Further, these Au-Ag hybrid nanosheets had high conductivity without reduced transparency. Therefore, this UV irradiation method, modified by adding metal ions is quite effective at improving and diversifying properties of Au nanosheets.

B) We used only AuNPs in this study, but we didn’t use AgNPs for preparing Au-Ag nanosheets.

Comment 2) What was the motivation for using Ag? Why two different Ag solutions - nitrate and acetate? The authors should clearly explain and describe the motivation for this work, what was obtained and how this is of interest.
Answer: Thank you for your comment.
We used silver as an example, i.e., silver was used as an example as it has a greater ionization tendency than gold. If we can demonstrate that the composition of gold nanosheets can be easily changed by metal species in solutions, a variety of nanosheets can be prepared from a single type of metal nanoparticle.

Introduction was changed as follows:

Original: To further develop this UV irradiation technique, a strategy is needed to produce nanosheets with various thicknesses and compositions from a single type of metal nanoparticle monolayer.

Revised: Furthermore, to change the composition of the nanosheet, it is necessary to synthesize NPs with the desired composition.An alternative is to synthesize two NPs with different metals and arrange them uniformly on water. Both methods, however, require considerable effort. Therefore, to further improve this UV irradiation technique, a strategy is needed to produce nanosheets with various thicknesses and compositions from a single type of metal nanoparticle monolayer…………..

In addition, we investigated whether the composition of Au nanosheets can be controlled by changing the metal species in solution. Here, silver, which alloys with gold and has a larger ionization tendency than gold, was selected. The silver electrolytes used were CH3COOAg, an inorganic ion, and AgNO3, an organic ion, as an anion.

Comment 3) In the results we can see that the authors varied the solution concentrations, yet this is not described at all in the experimental part. This should be changed and the experimental part needs to contain all experimental data.

Answer: Thank you for your advice. The following sentence was added in the experimental part.

Here, concentrations of 4.8, 48 and 480 µM were used for aqueous solutions of HAuCl4 and 5, 50 and 500 µM for aqueous solutions of AgNO3 and CH3COOAg.”

Comment 4) A) The solution concentrations for Au were 4.8, 48 and 480 and for silver acetate 5, 50 and 500, why not the same? Why did the authors choose these particular concentrations?

B) Why not vary the silver nitrate concentration too?
Answer: A) There is no particular meaning. This is due to the different concentrations of stock solutions in the lab, but both concentrations were made close together. We are sorry.
B) The morphology of nanosheets prepared on 500 µM (high concentration) silver nitrate was almost identical to that prepared on water, as shown in Fig. 7. Even when nanosheets were produced at low concentrations, the morphology remains unchanged (please see Word file). Thus, we did not describe the results of lower concentrations, however, the following sentences were added in section 3.2.2 in the revised manuscript.

On the other hand, the morphology of nanosheets prepared on 500 µM AgNO3 was almost identical to that prepared on water (Figure 7). Further, the appearance of nanosheets prepared on 4.8 µM and 48 µM AgNO3 was also uniform and similar to that prepared on water, indicating that the morphology in the AgNO3 system was independent of the concentration of AgNO3.”

Further, section name was changed from “Effect of CH3COOAg concentration” to “Effect of CH3COOAg and AgNOconcentrations”.

Comment 5) Why did the mesh size vary for Au? This has not been explained clearly. Didi it vary for Au also?
Answer: Thank you for your good question, but we don’t have the correct answer.
Considering only gold crystals, the spherical form is energetically most stable. However, on the water surface, the surface tension (surface energy) of the water, and the interface energy between the water and gold must also be considered. In system with a high amount of Au (500 µM), the cohesive force between gold would be predominant and Au product would be thicker. However, we have no evidence for this speculation and it is a guess.

Reviewer 3 Report

Comments and Suggestions for Authors

In this study, Tagawa et al., demonstrate a method to prepare highly transparent and conductive nanosheets using UV irradiation. Previously, they had shown that UV irradiation of gold nanoparticle (AuNP) monolayers on water resulted in transparent, conductive Au nanosheets. In this study, they extend this technique by using subphases containing metal ions.The researchers found that aqueous solutions containing gold (Au) and silver (Ag) ions were effective in modifying the Au nanosheets. When HAuCl4 solutions were used, the thickness of the Au nanosheets increased, which improved conductivity but decreased transparency. On the other hand, using aqueous AgNO3 and CH3COOAg solutions resulted in Au-Ag hybrid nanosheets. The morphology of these hybrid nanosheets depended on the electrolytes used. They were able to produce homogeneous Au-Ag nanosheets from AgNO3, as well as AuNPs covered with an Ag nanosheet from CH3COOAg.Importantly, these Au-Ag hybrid nanosheets exhibited high conductivity without reduced transparency. Therefore, the modified UV irradiation method, involving the addition of metal ions, proved to be effective in improving and diversifying the properties of the Au nanosheets. This research has implications for advanced optoelectronic applications where highly transparent and conductive nanosheets are desirable. Below are some minor comments. 

1.       What kind of support was utilized to prepare the gold nanosheet? To make it more clear the transparency, it is vital to add some photograph along with the control.  

2.       How does the concentration of HAuCl4 in the aqueous solution affect the size of the mesh-like Au nanosheets?

3.       How does the increase in film thickness affect the electrical resistances of the Au nanosheets?

4.       What are the electrical resistances of the Au nanosheets prepared on water and different HAuCl4 solutions?

5.       What is the significance of being able to prepare Au nanosheets using a single type of AuNP?

6.       Can author demonstrate if this type of synthesis is applicable to flexible optoelectronic?

Comments on the Quality of English Language

Not Required

Author Response

Thank you for your review.

Comment 1) A) What kind of support was utilized to prepare the gold nanosheet? B) To make it more clear the transparency, it is vital to add some photograph along with the control.
Answer: Thank you for your advice. A) Au nanosheets were prepared on water or aqueous solutions of HAuCl4 (please see 2.3 section “preparation of Au and Au-Ag nanosheets” ).
B) Photographs of Au nanosheets deposited on quartz plates were added in Figure 6. And Photographs of Au-Ag nanosheets were also added in Figure 12.

Comment 2) How does the concentration of HAuCl4 in the aqueous solution affect the size of the mesh-like Au nanosheets?
Answer: The size of the mesh-like Au nanosheets increased with increasing HAuCl4 concentrations. The following paragraph was added in second paragraph of section 3.1 preparation of thick Au nanosheets (Just below Figure 3).

TEM and SEM images (Figures 2 and 3) indicate that the amount of Au in these nanosheets increases with increasing concentrations of HAuCl4. The mass of each Au nanosheet was then measured using the QCM technique to evaluate how much the mass increased. The evaluated masses for water, 48 and 480 µM HAuCl4 samples were 1.33, 4.09 and 13.8 µg/cm2, respectively. Based on these masses, corresponding nanosheet thicknesses were calculated as 3.5, 10.8 and 36.5 nm, assuming a uniform gold film and gold density of 19.32 g/cm3. The increase in mass indicates that beside AuNPs on water, Au atoms produced by reduction of AuCl4 ions are the source of the Au nanosheet. Accordingly, Au nanosheets on HAuCl4 solutions are formed by fusion of AuNPs and deposition of Au atoms. Here, reduction of AuCl4 ions may occur near the AuNP monolayer, because the color of HAuCl4 solutions did not change after UV irradiation, remaining the original pale yellow. Further, UV irradiation of aqueous HAuCl4 solutions for an additional 60 min without an AuNP monolayer caused no change in the solution, meaning that AuCl4 ions are not reduced by UV irradiation alone.”

Comment 3) How does the increase in film thickness affect the electrical resistances of the Au nanosheets?
Answer: It is presumed that the thicker the film, the smaller the defects (disconnected portions etc) of mesh-like structure, and therefore the lower the resistance value.

Comment 4) What are the electrical resistances of the Au nanosheets prepared on water and different HAuCl4 solutions?

Answer: Thank you for your comment. The sentence was changed in section 3.1 preparation of thick Au nanosheets. Lines 166-167.

Original: Electrical resistances of Au nanosheets prepared on water, 48 µM and 480 µM HAuCl4 solutions were 2600 Ω/sq, 6.5 Ω/sq and 4.1 Ω/sq, respectively.
Revised: Electrical resistances of Au nanosheets prepared on water, 4.8 µM, 48 µM and 480 µM HAuCl4 solutions were 2600 Ω/sq, 2220Ω/sq, 6.5 Ω/sq and 4.1 Ω/sq, respectively.

Comment 5) What is the significance of being able to prepare Au nanosheets using a single type of AuNP?
Answer: Nanosheets with different thicknesses and different Au-Ag compositions can be produced without the cumbersome preparation of AuNP dispersions. Further, although this method is not applicable to large AuNPs, thick films can also be fabricated, it has the advantage that thick films can be fabricated by using solutions of metal ions. 
The following sentences have been written in Introduction of the original manuscript.

 “On the other hand, AuNPs with diameters 15 nm or less gave rise to flat nanosheets of uniform thickness, indicating that fusion between AuNPs was sufficient. Accordingly, this UV irradiation technique appears to be inapplicable for larger AuNPs. Nanosheets made from these larger particles exhibit high mechanical strength and conductivity, …”

Comment 6) Can author demonstrate if this type of synthesis is applicable to flexible optoelectronic?
Answer: Yes, we believe it is possible. The electrical resistance was measured after bending as shown in the attached photo (please see Word file), and the resistance was almost constant, although more detailed experiments are needed to describe the flexibility results in the paper.

Round 2

Reviewer 2 Report

Comments and Suggestions for Authors

The authors have addressed all questions raised in the review and implemented them in their revised paper. Therefore I recommend this work for publication in its present form.